# EMERGENCE OF CONSENSUS IN MULTI-AGENT Q-LEARNING FOR STOCHASTIC GAMES

## ABSTRACT

Multi-agent reinforcement learning (MARL) has made significant progress in diverse fields. A key challenge in MARL is consensus, which aligns individual estimates, reduces non-stationarity, and promotes coordinated behavior among agents. In this paper, we study a MARL system where agents interact in stochastic games and adapt their values and policies through independent Q-learning. In contrast to the prevailing literature that requires explicit consensus protocols, we study how consensus can emerge intrinsically without assuming any external coordination. We find that the covariance between Q-values and temporal-difference (TD) targets is the key quantity governing consensus, and the dynamics of variance of Q-values directly correspond to the second-order Price equation in evolutionary game theory. In addition, we prove that for large-scale anonymous stochastic games and a large batch size limit, independent learners naturally achieve consensus. We validate our findings through extensive agent-based simulations. Our results provide new insights into the learning dynamics of large-scale MARL systems, reveal the potential of intrinsic consensus to advance both theory and practice, and pave the way toward more scalable and efficient intelligent systems.

## 1 INTRODUCTION

One central challenge in multi-agent systems (MAS) is to design learning algorithms that enable agents to learn and collaborate efficiently. Multi-agent reinforcement learning (MARL) has recently made notable progress towards this goal. Building on advances in single-agent reinforcement learning (RL), the centralized training with decentralized execution (CTDE) paradigm has been proposed to enable agents to learn collaboratively (Lowe et al., 2017). Although CTDE could mitigate the non-stationarity of the environment, it still suffers from the credit assignment issue and limited scalability (Hernandez-Leal et al., 2019). Moreover, CTDE implicitly assumes that agents are fully cooperative, which does not always hold in real-world MAS (Kong et al., 2024).

On the other hand, in decentralized settings, agents pursue their own optimization objectives and learn independently. This approach scales naturally and is well-suited to large-scale systems such as swarm robotics (Brambilla et al., 2013). Since there is no central controller, decentralized learning brings the challenge of coordination, as it can be hard to align agents' behaviors without explicit communication. To address this, consensus mechanisms are often embedded into the learning process to enforce agreement on local estimates (Kar et al., 2013; Zhang et al., 2018). By equipping agents with explicit communication or information-sharing mechanisms to align their value estimates or policies during training, these methods can mitigate non-stationarity and help agents converge to cooperative strategies (Hernandez-Leal et al., 2017; Zhang & Zavlanos, 2019). Accordingly, frequent inter-agent communication is required, and algorithms often assume that all agents broadcast local learning signals or value function estimates to neighbors. The extra steps incur significant overhead that grows with the number of agents (Chen et al., 2022), which becomes costly in large-scale systems. An important question inevitably arises: whether explicit communication is truly necessary for agents to coordinate effectively, or equivalently, whether consensus at the system level can emerge inherently.

To examine the possible emergence behaviors without any external manipulation, it is necessary to analyze the learning dynamics of the system (Barfuss et al., 2025). Early studies drew parallels

between RL and evolutionary game theory (Börgers & Sarin, 1997; Tuyls et al., 2003), establishing the connection between RL algorithms and replicator dynamics (Barfuss et al., 2019). Subsequent works extended the analysis from individual to population-level dynamics, focusing not on the microscopic behavior of a particular agent but the macroscopic behavior of the system. For instance, Hu et al. (2019) derived the dynamics of multi-agent Q-learning in normal-form games, and Chu et al. (2023) generalized their results to stochastic games. These mean-field or continuum models capture learning trends observed in agent-based simulations and predict outcomes such as spontaneous cooperation in social dilemmas, yet the underlying difference or differential equations are rarely solvable in closed form. As a result, while they provide valuable qualitative insights, theoretical approaches typically rely on numerical integration, which can be computationally expensive when characterizing complex MARL dynamics. Meanwhile, regarding convergence, Leslie & Collins (2005); Kianercy & Galstyan (2012) investigate Q-learning in two-player normal-form games, and Hu et al. (2023) analyzed fictitious play in a multi-population structure, though still within relatively simple environments.

In this paper, we develop a deeper understanding of the emergence of consensus for independent Q-learning in stochastic games (Shapley, 1953) by analyzing the system-level learning dynamics. We focus on anonymous stochastic games, where rewards and transition dynamics depend solely on aggregate behavior rather than on agent identities (Yang et al., 2018; Guo et al., 2019). This setting captures large systems such as swarms or populations, where individual influence is negligible and aggregate effects dominate. At the individual level, the Q-value of a representative agent evolves according to a set of ordinary differential equations. At the population level, the distribution of Q-values follows a Fokker-Planck equation. Building upon these equations, macroscopic moments of the Q-value distribution, in particular the variance, evolve according to dynamics closely related to the second-order Price equation from evolutionary game theory (Price, 1970; 1972). Within the framework, the covariance between Q-values and temporal-difference targets directly governs the emergence of consensus. For normal-form games or under myopic agents, the analysis yields a direct variance-contraction condition that guarantees consensus. For general stochastic games, we provide a formal proof of consensus and identify how the learning rate $\alpha$ and discount factor $\gamma$ determine the convergence rate. We validate these theoretical predictions through extensive agent-based simulations. In summary, our work offers a theoretical foundation for understanding the emergent behavior in large-scale MARL systems and provides guidance for designing more scalable and efficient learning algorithms.

## 2 PRELIMINARIES

We consider a population of agents interacting in stochastic games and adapting their value estimates and policies via Q-learning (Watkins & Dayan, 1992).

### 2.1 STOCHASTIC GAMES

**Definition 2.1.** A *stochastic game* $\mathcal{G}$ is defined by a tuple $(\mathcal{N}, \mathcal{S}, \mathcal{A}, T, R)$, where

- $\mathcal{N} = \{1, 2, \cdots, N\}$ is the set of agents,

- $\mathcal{S} = \{s_1, s_2, \cdots, s_K\}$ is the set of environmental states,

- $\mathcal{A} = \{a_1, a_2, \cdots, a_M\}$ is the set of available actions in each state,

- $T : \mathcal{S} \times \mathcal{A}^N \times \mathcal{S} \to [0, 1]$ is the transition kernel, where $T(s, \boldsymbol{a}, s')$ specifies the probability of transitioning from state $s$ to state $s'$ when agents take joint action $\boldsymbol{a} = (a^1, a^2, \cdots, a^N)$,

- $R : \mathcal{N} \times \mathcal{S} \times \mathcal{A}^N \times \mathcal{S} \to \mathbb{R}$ is the reward function, where $R(i, s, \boldsymbol{a}, s')$ specifies the reward for agent $i$ when agents take joint action $\boldsymbol{a} = (a^1, a^2, \cdots, a^N)$ in state $s$ and transition to state $s'$.

Throughout this paper, superscripts denote indices of agents (e.g., $a^i$ is the action of agent $i$), while subscripts indicate indices of elements within the state or action space (e.g., $s_k$ is the $k$-th state and $a_m$ is the $m$-th action). For homogeneous large populations, it is natural to assume that tran-

sitions and rewards are independent of agent identities, a property formally known as *anonymity* in stochastic games (Adlakha et al., 2015).

**Definition 2.2.** A stochastic game $\mathcal{G}$ is *anonymous* if, for any states $s, s' \in \mathcal{S}$, joint action $\boldsymbol{a} \in \mathcal{A}^N$, and any permutation $\sigma(\boldsymbol{a}) := \left(a^{\sigma(1)}, a^{\sigma(2)}, \ldots, a^{\sigma(N)}\right)$, the following identities hold:

$$T(s, \boldsymbol{a}, s') = T\left(s, \sigma(\boldsymbol{a}), s'\right), \qquad R\left(i, s, \boldsymbol{a}, s'\right) = R\left(\sigma(i), s, \sigma(\boldsymbol{a}), s'\right). \tag{1}$$

A direct consequence of anonymity is that transitions and rewards depend only on the distribution of actions, not on the identities of the agents taking them.

**Lemma 2.3.** *In anonymous stochastic games, the transition kernel can be represented as $\tilde{T}(s, \boldsymbol{x}, s')$, where $\boldsymbol{x}$ is the $M$-dimensional vector specifying the proportion of agents choosing each action. Similarly, the reward function can be expressed as $\tilde{R}(a|s, \boldsymbol{x}, s')$, representing the reward of an agent that takes action $a$ in state $s$ and transitions to state $s'$ when the population action distribution is $\boldsymbol{x}$.*

### 2.2 Q-LEARNING

We further assume that agents update their policies using Q-learning (Watkins & Dayan, 1992). Each agent $i$ maintains a Q-value vector $\boldsymbol{Q}(i) = \left(Q(i, s_1, a_1), Q(i, s_1, a_2), \cdots, Q(i, s_K, a_M)\right)$. Actions are selected according to the softmax policy:

$$X(i, s, a) = \frac{\exp(Q(i, s, a))}{\sum_{a'} \exp(Q(i, s, a'))}. \tag{2}$$

A batch learning framework is adopted with batch size $B$. During each interaction phase, agent $i$ samples actions according to equation 2, interacts with the environment, observes the next state, and receives the reward. This process is repeated $B$ times, yielding a dataset $\{(s, a(i), r(i), s')\}$ used to update Q-values. For each sample, its temporal-difference (TD) target is computed as

$$r(i) + \gamma \max_{a'} Q(i, s', a'), \tag{3}$$

where $\gamma$ is the discount factor. The TD targets are then aggregated over $(s, a)$-pairs and averaged to obtain $D(i, s, a)$, which updates the Q-values via

$$Q(i, s, a) \leftarrow (1 - \alpha)Q(i, s, a) + \alpha D(i, s, a), \tag{4}$$

where $\alpha$ denotes the learning rate.

## 3 LEARNING DYNAMICS OF THE SYSTEM

The MARL system is inherently stochastic: each agent selects actions at random based on its policies, while the environment evolves according to a stochastic transition kernel. Henceforth, even when the macroscopic dynamics exhibit a clear trend toward consensus, persistent fluctuations prevent exact consensus. To characterize the consensus dynamics, we adopt the large-batch regime $B \to \infty$, in which empirical averages converge to expectations by the law of large numbers. In this limit, sampling noise in the TD targets vanishes, yielding deterministic learning dynamics of the system (Barfuss et al., 2019). For completeness, the appendix also presents an analysis of the learning dynamics without batch learning, using a stochastic differential equation (SDE) framework.

### 3.1 DYNAMICS OF Q-VALUES

Under the large-batch regime, the difference equations provide an exact description of the learning dynamics. However, it becomes difficult to analyze the macroscopic behavior of a large population through discrete dynamics. To bridge individual-level and population-level descriptions, we adopt an ordinary differential equation (ODE) formulation. The evolution of each agent's Q-value is captured by

$$\frac{\mathrm{d}}{\mathrm{d}t}Q(i, s, a) = \mu(i, s, a). \tag{5}$$

Here, $\mu(i, s, a)$ represents the expected update of agent $i$'s Q-value for action $a$ in state $s$. It is computed as the learning rate multiplied by the TD error, averaged over the opponents' joint action $\boldsymbol{a}^{-i}$ and the next state $s'$:

$$
\begin{aligned}
\mu(i, s, a) &= \alpha \mathbb{E}[r(i) + \gamma \max_{a'} Q(i, s', a') - Q(i, s, a) | s, a] \\
&= \alpha \Big[ \sum_{\boldsymbol{a}^{-i}} \Big[ \prod_{j \neq i} X(j, s, a^j) \Big] \sum_{s'} T(s, a + \boldsymbol{a}^{-i}, s') D(i, s, a | \boldsymbol{a}^{-i}, s') - Q(i, s, a) \Big],
\end{aligned}
\tag{6}
$$

where $D(i, s, a | \boldsymbol{a}^{-i}, s')$ is the TD error for agent $i$ when it takes action $a$, its opponents take joint action $\boldsymbol{a}^{-i}$ in state $s$, and the next state is $s'$:

$$
D(i, s, a | \boldsymbol{a}^{-i}, s') = R(i, s, a + \boldsymbol{a}^{-i}, s') + \gamma \max_{a'} Q(i, s', a').
\tag{7}
$$

The ODE equation 5 describes the evolution of each agent's Q-values. For small populations, one can obtain analytical insights, including equilibrium points, robustness, and resilience, by applying dynamical systems theory (Kianercy & Galstyan, 2012; Bloembergen et al., 2015; Barfuss et al., 2019). As the population size grows, tracking the behavior of each agent quickly becomes intractable. To understand how collective outcomes emerge from individual learning and how consensus in Q-values and policies arises in large systems, we move from agent-specific descriptions to population-level characterizations.

**Lemma 3.1.** *For a population of agents taking a joint policy $\boldsymbol{X}$ in an anonymous stochastic game $\mathcal{G}$, the conditional reward $\mathbb{E}[r(i)|s, a]$ of agent $i$ when taking action $a$ in state $s$ becomes independent of the index $i$ as the population size $N \to \infty$.*

*Proof.* For an arbitrary agent $i$, we have

$$
\begin{aligned}
\mathbb{E}[r(i)|s, a] &= \mathbb{E}_{\boldsymbol{a}^{-i}, s'} \big[ R(i, s, a + \boldsymbol{a}^{-i}, s') \big] \\
&= \mathbb{E}_{\boldsymbol{x}^{-i}, s'} \big[ \tilde{R}(a, s, \boldsymbol{x}(a + \boldsymbol{a}^{-i}), s') \big] \\
&\xrightarrow{N \to \infty} \mathbb{E}_{s'} \big[ \tilde{R}(a, s, \bar{\boldsymbol{X}}(s), s') \big],
\end{aligned}
\tag{8}
$$

where $\boldsymbol{x}^{-i}$ denotes the proportion of each action taken by all agents except $i$, and $\bar{\boldsymbol{X}}(s)$ is the mean policy of the system in state $s$, whose $a$-component is computed as

$$
\bar{\boldsymbol{X}}(s, a) = \frac{1}{N} \sum_{i=1}^{N} X(i, s, a).
\tag{9}
$$

The last equality follows from the law of large numbers: as $N \to \infty$, the influence of a single agent's action on $\boldsymbol{x}$ vanishes, and therefore $\boldsymbol{x}^{-i}$ almost surely converges to the mean policy $\bar{\boldsymbol{X}}(s)$. $\square$

The significance of this lemma is twofold. First, it allows us to describe agents in terms of their Q-values and policies rather than their labels, thereby simplifying the analytical model. The notations are updated accordingly. For example, the policy of an arbitrary agent with Q-value $\boldsymbol{Q}$ is now written as $\boldsymbol{X}(\boldsymbol{Q})$ with component $X(\boldsymbol{Q}, s, a)$, replacing $\boldsymbol{X}(i)$ and $X(i, s, a)$. And for convenience, the $(s, a)$-component of the Q-value is further simplified from $Q(\boldsymbol{Q}, s, a)$ to $Q(s, a)$. Second, since the expected immediate reward of an agent is independent of the index, it is also independent of its Q-value. This property is crucial for establishing consensus in the system.

As shown in the proof of Lemma 3.1, both action distributions $\boldsymbol{x}$ and $\boldsymbol{x}^{-i}$ converge almost surely to the system's mean policy $\bar{\boldsymbol{X}}(s)$ in the limit of large populations. We can then leverage a mean-field approximation to simplify the dynamics. Specifically, instead of interacting with individual agents taking heterogeneous policies, each agent can be regarded as interacting with a virtual agent that follows the average policy of the entire population. Consequently, the Q-value dynamics for large-scale systems reduce to

$$
\frac{\mathrm{d}}{\mathrm{d}t} Q(s, a) = \alpha [\bar{D}(\boldsymbol{Q}, s, a) - Q(s, a)].
\tag{10}
$$

Here, $\bar{D}(\boldsymbol{Q}, s, a)$ denotes the TD error for a representative agent with Q-value $\boldsymbol{Q}$ when taking action $a$ in state $s$:

$$
\bar{D}(\boldsymbol{Q}, s, a) = \sum_{s'} T(s, \bar{\boldsymbol{X}}, s') \Big[ \tilde{R}(a, s, \bar{\boldsymbol{X}}, s') + \gamma \max_{a'} Q(s', a') \Big].
\tag{11}
$$

In fact, the learning dynamics of such large-scale systems can be interpreted as a flow in the Q-value space, where the velocity field at each point is given by the ODE equation 10. The corresponding evolution of the probability distribution over Q-values is governed by the diffusion-free Fokker-Planck equation (FPE):

$$\frac{\partial p_t(Q)}{\partial t} = -\nabla_{Q(s,a)} \cdot \Big[ \boldsymbol{\mu}(\boldsymbol{Q}) p_t(Q) \Big],$$

(12)

where $\nabla_{Q(s,a)}$ denotes the gradient with respect to $Q(s,a)$. The term $\boldsymbol{\mu}(\boldsymbol{Q}) p_t(\boldsymbol{Q})$ represents the flux in Q-value space, and its divergence measures the local outflow or inflow of probability mass. Intuitively, regions with positive divergence act as sources, pushing probability mass outward, while regions with negative divergence act as sinks, pulling probability inward. In the large-batch limit considered here, the dynamics are purely advective. For finite batch sizes, a diffusion term would appear in the FPE:

$$\tfrac{1}{2} \nabla_Q \cdot \Big\{ \nabla_Q \cdot \Big[ \boldsymbol{\Sigma}(\boldsymbol{Q}) \, p_t(\boldsymbol{Q}) \Big] \Big\},$$

(13)

where $\boldsymbol{\Sigma}(\boldsymbol{Q})$ is the covariance matrix of the Q-values.

## 4  DYNAMICS OF VARIANCE OF Q-VALUES

In brief, equation 10 and equation 12 describe the system dynamics at the individual and population levels, respectively. However, they do not directly reveal the evolutionary outcomes or the conditions under which consensus emerges. To bridge this gap, we investigate the macroscopic statistics that summarize population-level behaviors. In particular, we pay attention to the variance of Q-values across agents, which serves as an informative measure of alignment versus dispersion. After introducing the relevant moment-related quantities, we derive the evolution equation governing the variance of Q-values.

**Definition 4.1.** We define the mean and variance of the $(s,a)$-component of a random variable $Y$ across all agents at time $t$ as

$$\mathbb{E}_t[Y] = \int Y p_t(\boldsymbol{Q}) \mathrm{d}\boldsymbol{Q},$$

(14)

$$\mathrm{Var}_t[Y] = \int (Y - \mathbb{E}_t[Y])^2 p_t(\boldsymbol{Q}) \mathrm{d}\boldsymbol{Q}.$$

(15)

**Theorem 4.2.** In the large-batch limit $B \to \infty$, the variance of the $(s,a)$-component of $\boldsymbol{Q}$ evolves according to

$$\frac{\partial}{\partial t} \mathrm{Var}_t[Q(s,a)] = 2\alpha \Big\{ \mathrm{Cov}_t[Q(s,a), \bar{D}_t(\boldsymbol{Q}, s, a)] - \mathrm{Var}_t[Q(s,a)] \Big\},$$

(16)

where the covariance $\mathrm{Cov}_t$ between $Q(s,a)$ and the corresponding average TD target $\bar{D}_t(\boldsymbol{Q}, s, a)$ is defined by

$$\mathrm{Cov}_t\big[Q(s,a), \bar{D}_t(\boldsymbol{Q}, s, a)\big]$$
$$= \int \big(Q(s,a) - \mathbb{E}_t[Q(s,a)]\big)\big(\bar{D}_t(\boldsymbol{Q}, s, a) - \mathbb{E}_t[\bar{D}_t(\boldsymbol{Q}, s, a)]\big) p_t(\boldsymbol{Q}) \mathrm{d}\boldsymbol{Q}.$$

(17)

*Proof.* For $B \to \infty$, the individual-level dynamics of Q-values satisfy the ODE equation 10. The system-level dynamics of the first moment (mean Q-value) can be obtained by differentiating $\mathbb{E}_t[Q(s,a)]$:

$$\frac{\partial}{\partial t} \mathbb{E}_t[Q(s,a)] = \mathbb{E}[\frac{\partial}{\partial t} Q(s,a)] = \alpha \Big( \mathbb{E}[\bar{D}_t(\boldsymbol{Q}, s, a)] - \mathbb{E}_t[Q(s,a)] \Big).$$

(18)

Similarly, the dynamics of the second moment are

$$\frac{\partial}{\partial t} \mathbb{E}_t[Q(s,a)^2] = 2\alpha \Big\{ \mathbb{E}_t[Q(s,a) \bar{D}_t(\boldsymbol{Q}, s, a)] - \mathbb{E}_t[Q(s,a)^2] \Big\}.$$

(19)

Since $\mathrm{Var}_t[Q(s,a)] = \mathbb{E}_t[Q(s,a)^2] - \mathbb{E}_t[Q(s,a)]^2$, its time derivative can be written as

$$\frac{\partial}{\partial t} \mathrm{Var}_t[Q(s,a)] = 2\mathrm{Cov}_t\big[Q(s,a), \frac{\partial}{\partial t} Q(s,a)\big].$$

(20)

By substituting equation 18 and equation 19 into equation 20, we immediately get the desired result. $\square$

It is noteworthy that the variance dynamics can be seen as a natural generalization of the second-order Price equation in our MARL setting (Price, 1972), which characterizes how population trait diversity evolves in evolutionary game theory. In that context, if individuals with higher trait values tend to achieve higher fitness, the diversity of that trait also tends to increase. By interpreting the TD target as the "fitness" of an agent in the environment, a clear analogy emerges: agents with higher Q-values that tend to achieve larger TD targets amplify variance, whereas weaker correlations suppress variance and promote consensus. The additional term $-2\alpha \text{Var}_t(Q)$ reflects a mean-regression effect intrinsic to Q-learning updates, which counteracts divergence and stabilizes the dynamics.

**Corollary 4.3.** *For stateless games $|\mathcal{S}| = 1$ or for myopic agents $\gamma = 0$, the variance dynamics in equation 16 reduces to*

$$\frac{\partial}{\partial t}\text{Var}_t[Q(s,a)] = -2\alpha \cdot \text{Var}_t[Q(s,a)], \tag{21}$$

*which admits the closed-form solution*

$$\text{Var}_t[Q(s,a)] = e^{-2\alpha t} \cdot \text{Var}_0[Q(s,a)]. \tag{22}$$

*Thus, the variance of each action decays exponentially at rate $2\alpha$, and the Q-values of all agents converge to a common equilibrium.*

*Proof.* In both cases, the expected immediate reward term in $\bar{D}_t(\boldsymbol{Q}, s, a)$ is independent of $\boldsymbol{Q}$. For stateless games, the bootstrap term $\max_{a'} Q(s', a')$ cancels out in the covariance. For myopic agents, the bootstrap term is absent altogether. Hence, in both cases, the covariance term vanishes. $\square$

## 5 CONSENSUS IN STOCHASTIC GAMES

To establish consensus of Q-values in general settings, we leverage the Lipschitz property of the average TD target $\bar{D}_t(\boldsymbol{Q}, s, a)$ and show that the $\ell_\infty$ norm of Q-value differences decays to zero exponentially.

**Lemma 5.1.** *For any $\boldsymbol{Q}$, $\boldsymbol{Q}'$, and fixed $(s, a)$, $\bar{D}_t$ is nondecreasing in each coordinate $Q(s', a')$ and satisfies the Lipschitz condition*

$$\left|\bar{D}_t(\boldsymbol{Q}, s, a) - \bar{D}_t(\boldsymbol{Q}', s, a)\right| \leq \gamma \|\boldsymbol{Q} - \boldsymbol{Q}'\|_\infty. \tag{23}$$

*Moreover, if only a single coordinate changes, then the Lipschitz constant along that coordinate is at most the discount factor $\gamma$.*

*Proof.* It is clear that the immediate reward term is independent of Q-values. For the future reward term, notice that the map $\boldsymbol{x} \mapsto \max_{a'} x_{a'}$ is 1-Lipschitz with respect to the $\ell_\infty$ norm. Averaging over the joint action profile $\boldsymbol{a}$ and the transition kernel $T$ yields a convex combination, which preserves the 1-Lipschitz property. Multiplying by $\gamma$ in the future reward term gives equation 23. Monotonicity follows because increasing any $Q(s', a')$ cannot decrease $\max_{a''} Q(s', a'')$ or convex combinations of such maxima. The single-coordinate bound is a direct specialization. $\square$

**Theorem 5.2.** *For any two agents with Q-values $\boldsymbol{Q}$ and $\boldsymbol{Q}'$, in the large-batch limit $B \to \infty$, the $\ell_\infty$ norm of their difference $\Delta_t = \|\boldsymbol{Q} - \boldsymbol{Q}'\|_\infty$ satisfies*

$$\frac{\partial \Delta_t}{\partial t} \leq -\alpha(1 - \gamma)\,\Delta_t, \tag{24}$$

*or equivalently,*

$$\Delta_t \leq \exp[-\alpha(1 - \gamma)\,t]\,\Delta_0. \tag{25}$$

*Proof.* The dynamics of the $(s, a)$-components of the two agents' Q-values are

$$\frac{\partial}{\partial t}Q(s,a) = \alpha\Big(\bar{D}_t(\boldsymbol{Q}, s, a) - Q(s, a)\Big), \qquad \frac{\partial}{\partial t}Q'(s,a) = \alpha\Big(\bar{D}_t(\boldsymbol{Q}', s, a) - Q'(s, a)\Big). \tag{26}$$

Subtracting these two equations yields

$$\frac{\partial}{\partial t}[Q(s,a) - Q'(s,a)] = \alpha\Big\{\big[\bar{D}_t(\boldsymbol{Q}, s, a) - \bar{D}_t(\boldsymbol{Q}', s, a)\big] - [Q(s, a) - Q'(s, a)]\Big\}. \tag{27}$$

Therefore, the time derivative of the absolute difference is

$$\frac{\partial}{\partial t}|Q(s,a) - Q'(s,a)|$$

$$= \text{sign}\big[Q(s,a) - Q'(s,a)\big] \cdot \frac{\partial}{\partial t}\big[Q(s,a) - Q'(s,a)\big]$$

$$= \alpha \Big\{ \text{sign}\big[Q(s,a) - Q'(s,a)\big] \cdot \big[\bar{D}_t(\boldsymbol{Q}, s, a) - \bar{D}_t(\boldsymbol{Q}', s, a)\big] - |Q(s,a) - Q'(s,a)| \Big\} \quad (28)$$

$$\leq \alpha \Big\{ |\bar{D}_t(\boldsymbol{Q}, s, a) - \bar{D}_t(\boldsymbol{Q}', s, a)| - |Q(s,a) - Q'(s,a)| \Big\}$$

$$\leq \alpha \Big\{ \gamma \|\boldsymbol{Q} - \boldsymbol{Q}'\|_\infty - |Q(s,a) - Q'(s,a)| \Big\} \text{ (Lemma 5.1).}$$

Since this inequality holds uniformly for all $(s,a)$, it also applies to the $\ell_\infty$ norm $\Delta_t$. □

As a consequence, all agents' Q-values converge to a common fixed point $Q^*$ satisfying

$$Q^*(s,a) = \bar{D}_t(\boldsymbol{Q}^*, s, a). \quad (29)$$

In the limit of large populations, this condition takes the form of

$$Q^*(s,a) = \sum_{s'} \tilde{T}(s, \boldsymbol{X}^*(s), s')\Big[\tilde{R}(s, \boldsymbol{X}^*(s), s') + \gamma \max_{a'} Q^*(s', a')\Big], \quad (30)$$

where $\boldsymbol{X}^*$ is the consensus policy induced by $Q^*$. Concretely, in equilibrium, not only do all agents' Q-values coincide, but each agent's policy is also optimal with respect to the consensus policy of the entire population.

# 6 EXPERIMENTS

To validate our theoretical results, we conduct extensive agent-based simulations on both normal-form and stochastic games.

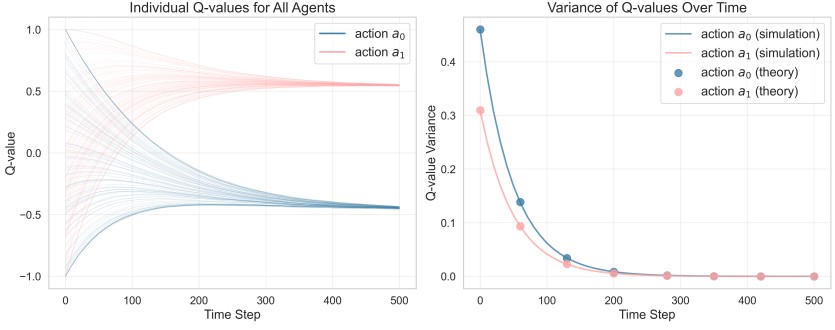

Figure 1: Evolution of individual Q-values and variance of Q-values.

## 6.1 NORMAL-FORM GAMES

We first consider the pairwise prisoner's dilemma game with a payoff matrix

$$\begin{array}{c|cc} & a_1 & a_2 \\ \hline a_1 & b-c & -c \\ a_2 & b & 0 \end{array}. \quad (31)$$

Here, agents can choose to cooperate ($a_1$) or defect ($a_2$). The entry $(a_i, a_j)$ gives the payoff of the row agent when choosing action $a_i$ against the column agent choosing $a_j$. At each time step, agents select their actions, interact pairwise, and obtain the average rewards from these interactions, which are used to update their Q-values. We set the parameters as $N = 100$, $b = 2$, $c = 1$, $\alpha = 0.01$, $\beta = 1$, and $\gamma = 0.9$.

The left panel of Figure 1 shows that, despite heterogeneous initializations, the Q-values associated with each action rapidly align and converge to a common equilibrium. The right panel compares the simulated variance of Q-values with the theoretical variance, namely the exponential decay predicted by Corollary 4.3. The close agreement between empirical trajectories and the theoretical curves confirms both consensus formation and the predicted convergence rate.

## 6.2 STOCHASTIC GAMES

We next consider the Ecological Public Goods (EPG) game introduced in (Barfuss et al., 2020). The environment alternates between a prosperous state $s_1$ and a degraded state $s_2$. In $s_1$, agents engage in a standard Public Goods Game: choosing cooperation ($a_1$) incurs an individual cost $c$ and contributes $c \times r$ to a collective reward pool, which is then evenly distributed among all agents. On the other hand, defection increases the risk of state degradation: with probability $p_{12} \cdot n_D/N$, where $n_D$ is the number of defectors in the current round, the environment transitions to $s_2$. In this degraded state, all agents receive the same reward $m$. And with probability $p_{21} \cdot n_C/N$, where $n_C$ is the number of cooperators in the current round, the environment returns from $s_2$ to $s_1$. We set the parameters as $N = 100$, $r = 2$, $c = 1$, $m = 0$, $p_{12} = p_{21} = 0.8$, $\alpha = 0.01$, $\beta = 1$, and $\gamma = 0.5, 0.7, 0.9$.

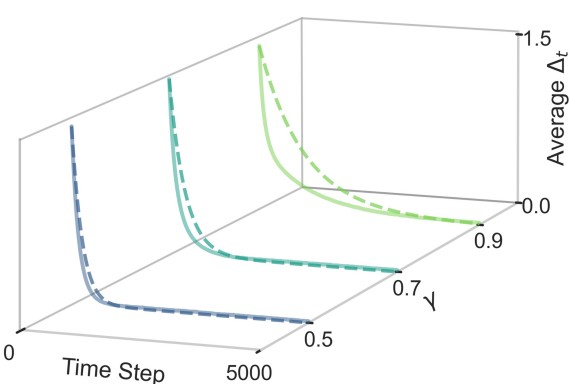

Figure 2: Evolution of the average difference of Q-values $\Delta_t$ under varying discount factors $\gamma$. The dashed lines indicate the theoretical upper bounds, and the solid lines show the simulation results.

For the above dynamic environment, Figure 2 illustrates the evolution of the average difference $\Delta_t := \|Q - Q'\|_\infty$, comparing simulation results with theoretical upper bounds. The empirical curves decrease sharply at the initial stage and remain strictly below the theoretical trajectories computed using equation 25. Moreover, increasing the discount factor $\gamma$ slows down consensus among Q-values. This is expected: a higher $\gamma$ assigns more weight to the bootstrapped term $\gamma \max_{a'} Q(s', a')$, which propagates heterogeneity from the next state and thus prolongs transient disagreement across agents. Nevertheless, since the TD operator remains $\gamma$-Lipschitz for any $\gamma \in [0, 1)$, the contraction property is always preserved. Consequently, Q-values across agents still contract at the rate $\alpha(1 - \gamma)$, in the sense that a larger $\gamma$ increases the time required to reach consensus but does not alter the final outcome.

**Finite batch and finite population.** In the derivations, we isolate the intrinsic consensus trend by taking the large-batch and large-population limits. To assess the robustness of our theoretical predictions, we relax both assumptions and vary the batch size $B$ and the population size $N$. Figure 3 depicts the simulation trajectories of the average $\Delta_t$ (solid line) alongside the theoretical upper bound from equation 25 (dashed line).

In the left panel, with fixed $N = 100$ and $\gamma = 0.9$, smaller batch sizes introduce stronger fluctuations, yet the empirical trajectories remain strictly below the theoretical bound. From a dynamical-

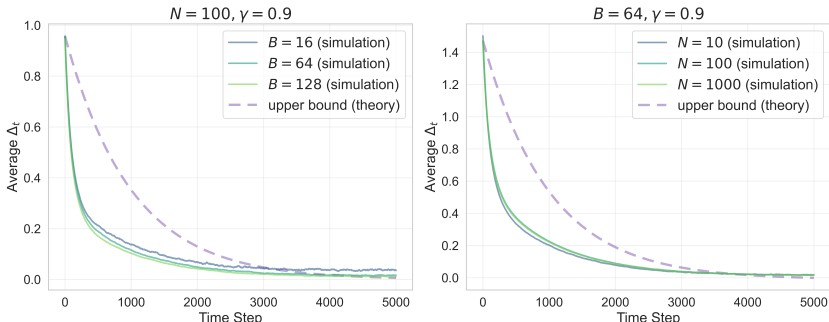

Figure 3: Evolution of the average difference $\Delta_t$ for varying batch sizes $B$ and population sizes $N$. The dashed lines indicate the theoretical upper bounds, and the solid lines show the simulation results.

systems perspective, the randomness induced by finite batch sizes adds a zero-mean diffusion term to the population-level evolution of Q-values. However, the contractive drift dominates, driving the system toward consensus at the exponential rate $\alpha(1 - \gamma)$ predicted by equation 25. In the right panel, we vary the population size $N$ from 10 to 1000. Although the independence of the expected immediate reward term from individual Q-values was formally established using the law of large numbers, this approximation remains accurate even for relatively small populations. Consequently, the consensus outcome is preserved across all tested parameter settings.

## 7  CONCLUSION AND FUTURE WORK

In this work, we study the emergent consensus behavior of independent Q-learning agents in stochastic games. Building upon the ODE and FPE characterizations of Q-value dynamics, we prove that agents can spontaneously align their estimates and reach consensus without any explicit coordination protocol, a phenomenon confirmed by extensive agent-based simulations.

Our first main finding is that the emergence of consensus depends closely on the covariance between Q-values and TD targets. The variance dynamics is formally analogous to the second-order Price equation: when the covariance is smaller than the variance of Q-values, consensus is guaranteed. Importantly, this result is not limited to anonymous stochastic games. A promising direction for future work is to relate the covariance to structural quantities of the system, such as the spectral norm of the adjacency matrix in networked games (Hussain et al., 2025), which could broaden the applicability of our theoretical results. Moreover, the analytical framework developed here can be readily extended to scenarios involving function approximation (Mnih et al., 2015) or alternative learning algorithms.

The second key contribution is the rigorous proof of consensus among agents' Q-values in stochastic games, from which we derived the equilibrium condition formalized as the consensus-based Bellman equation. Solving this equation yields the consensus equilibria of the population. Our findings complement and extend the mean-field RL framework (Yang et al., 2018; Laurière et al., 2022) by providing a dynamical perspective on how independent learners synchronize their Q-values. Furthermore, the homogeneity of Q-values at equilibrium enables us to transform the multi-agent stochastic game into a single-agent optimal control problem (Sutton et al., 2002), where transitions and rewards are governed by the population's homogeneous policy. The equilibrium solutions of this reduced problem coincide with the consensus equilibria of the original stochastic game. In this sense, our results also connect to self-play RL (Silver et al., 2017).

Finally, previous work (Barfuss et al., 2019) has shown that the learning dynamics of Boltzmann Q-learning incorporate an entropy regularization term, whereas vanilla A2C does not (Mnih et al., 2016). Since A2C lacks consensus guarantees, it is natural to ask whether the emergence of consensus in Q-learning is influenced by entropy regularization. Addressing this question through further theoretical analysis may provide deeper insights into the mechanisms underlying emergent consensus in multi-agent systems.

ETHICS STATEMENT

This work adheres to the ICLR Code of Ethics. In this study, no human subjects, datasets, or animal experimentation was involved. We have taken care to avoid any biases or discriminatory outcomes in our research process. No personally identifiable information was used, and no experiments were conducted that could raise privacy or security concerns. We are committed to maintaining transparency and integrity throughout the research process.

REPRODUCIBILITY STATEMENT

We have made every effort to ensure that the results presented in this paper are reproducible. All code has been uploaded in the Supplementary Material to facilitate replication and verification. The experimental setup, including training steps, model configurations, is described in detail in the paper. We believe these measures will enable other researchers to reproduce our work and further advance the field.

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

# A  APPENDIX

## A.1  USE OF LLMS

Large Language Models (LLMs) were used solely for grammar correction. It is important to note that the LLM was not involved in the ideation, research methodology, or experimental design. All research concepts, ideas, and analyses were developed and conducted by the authors. The contributions of the LLM were solely focused on improving the linguistic quality of the paper, with no involvement in the scientific content or data analysis.

The authors take full responsibility for the content of the manuscript, including any text generated or polished by the LLM. We have ensured that the LLM-generated text adheres to ethical guidelines and does not contribute to plagiarism or scientific misconduct.

## A.2  DYNAMICS OF Q-VALUES

For $B = 1$, the individual agent's Q-values and policy updates are stochastic. Therefore, we use a SDE to model the dynamics of a representative agent's Q-values:

$$d\boldsymbol{Q} = \boldsymbol{\mu}(\boldsymbol{Q})dt + \sqrt{\boldsymbol{\Sigma}(\boldsymbol{Q})}d\boldsymbol{W}. \tag{32}$$

Here, the drift term $\boldsymbol{\mu}(\boldsymbol{Q})$ still represents the average trend of individual Q-values update, and is computed by

$$\boldsymbol{\mu}(\boldsymbol{Q}, s, a) = \alpha X(\boldsymbol{Q}, s, a)p_t(s)\Big(\bar{D}_t(\boldsymbol{Q}, s, a) - Q(s, a)\Big). \tag{33}$$

Comparing with equation 10, the additional policy term $X(\boldsymbol{Q}, s, a)$ and state distribution term $p_t(s)$ appears in $\boldsymbol{\mu}$ because only the Q-value of the selected action in the current state is updated. On the other hand, the diffusion term $\boldsymbol{\Sigma}(\boldsymbol{Q})$ quantifies the stochasticity of the learning process, which is computed as the covariance matrix of Q-value changes in the current time step:

$$\boldsymbol{\Sigma}(\boldsymbol{Q}, (s, a), (s', a')) = \alpha^2 \mathbb{E}_t\big[[D_t(\boldsymbol{Q}, s, a) - Q(s, a)][D_t(\boldsymbol{Q}, s', a') - Q(s', a')]\big]. \tag{34}$$

The SDE describes the learning dynamics of Q-values at the individual level. At the population level, we use the corresponding FPE to describe the dynamics of the probability density $p_t(\boldsymbol{Q})$ of Q-values:

$$\frac{\partial p_t(\boldsymbol{Q})}{\partial t} = -\nabla_{\boldsymbol{Q}} \cdot \big[\boldsymbol{\mu}(\boldsymbol{Q})\, p_t(\boldsymbol{Q})\big] + \frac{1}{2}\nabla_{\boldsymbol{Q}} \cdot \Big\{\nabla_{\boldsymbol{Q}} \cdot \big[\boldsymbol{\Sigma}(\boldsymbol{Q})\, p_t(\boldsymbol{Q})\big]\Big\}. \tag{35}$$

## A.3  DYNAMICS OF STATE DISTRIBUTION

The time evolution of the probability $p_t(s)$ that the system is in state $s$ is the difference between the probability flux into and out of state $s$. Formally, without Q-values update, the state dynamics is represented by a master equation. With Q-values update, there is an additional $\mathcal{O}(\alpha)$ term:

$$\begin{aligned}
\frac{\partial p_t(s)}{\partial t} =\ & \sum_{s'}\sum_{\boldsymbol{a}} \Big[\prod_{a^i} X_{s'a^i}(\boldsymbol{Q})\Big] T(s', \boldsymbol{a}, s)p_t(s') \\
& - \sum_{s'}\sum_{\boldsymbol{a}} \Big[\prod_{a^i} X_{sa_i}(\boldsymbol{Q})\Big] T(s, \boldsymbol{a}, s')p_t(s) + \mathcal{O}(\alpha).
\end{aligned} \tag{36}$$

In the large-population limit, we can use mean-field approximation to simplify equation 36:

$$\frac{\partial p_t(s)}{\partial t} = \sum_{s'} \tilde{T}(s', \bar{\boldsymbol{X}}(s'), s) p_t(s')$$
$$- \sum_{s'} \tilde{T}(s, \bar{\boldsymbol{X}}(s), s') p_t(s) + \mathcal{O}(\alpha). \tag{37}$$

### A.4 TIME-SCALE SEPARATION

In the SDE and the FPE, the drift term is of order $\mathcal{O}(\alpha)$. It's easy to see from the definition of $\boldsymbol{\Sigma}(\boldsymbol{Q})$ that the diffusion term is of order $\mathcal{O}(\alpha^2)$. In addition, since the master equation is of order $\mathcal{O}(1)$ in $\alpha$, we obtain a time-scale separation under $\alpha \ll 1$. In such case, the state distribution evolves significantly faster than the Q-values.

**Theorem A.1.** *Under the time-scale separation $\alpha \ll 1$, the state distribution converges to $p^*(s)$ defined by the following solution of the following set of equations:*

$$p^*(s) = \sum_{s'} \tilde{T}(s', \bar{X}(s'), s) p^*(s'). \tag{38}$$

The state distribution $p^*(s)$ is exactly the stationary state distribution of the Markov chain induced by the transition kernel and the population policy profile $\boldsymbol{X}$. In the population limit, $p^*(s)$ only depends $\tilde{T}$ and the mean policy $\bar{X}$ of the system.

**Corollary A.2.** *Under the time-scale separation $\alpha \ll 1$, the population-level dynamics of Q-values is given by*

$$\frac{\partial p_t(\boldsymbol{Q})}{\partial t} = -\nabla_{\boldsymbol{Q}} \cdot \left[ \boldsymbol{\mu}_t^*(\boldsymbol{Q}) \, p_t(\boldsymbol{Q}) \right], \tag{39}$$

*where $\boldsymbol{\mu}_t^*(\boldsymbol{Q})$ is drift vector in which all the terms are evaluated at the stationary state distribution $p^*(s)$.*

### A.5 DYNAMICS OF VARIANCE

**Theorem A.3.** *In the limit $\alpha \ll 1$ and under $B = 1$, the variance of the $(s, a)$ component of $\boldsymbol{Q}$ evolves according to*

$$\frac{\partial}{\partial t} \text{Var}_t[Q(s, a)] = 2\alpha \Big\{ \text{Cov}_t\big(Q(s, a), p^*(s) X(\boldsymbol{Q}, s, a) \bar{D}_t^*(\boldsymbol{Q}, s, a)\big) - \text{Var}_t[Q(s, a)] \Big\}. \tag{40}$$

By following an analysis analogous to the large-batch setting, we find that the variance dynamics are governed by the covariance between $Q$-values and the rescaled TD targets, where the rescaling incorporates both the stationary state distribution and the action-selection probabilities of the policy. Taking the analogy with the second-order Price equation, in the non-batch learning regime ($B = 1$), the individual-level "fitness" along the $(s, a)$ dimension is precisely the TD target rescaled by the state distribution and the policy.

