# OpenReview forum: "Emergence of Consensus in Multi-Agent Q-learning for Stochastic Games"
_ICLR.cc/2026/Conference — Submitted to ICLR 2026_

### Official Review · Reviewer_aX3W · 2025-10-29

**Soundness:** 3
**Presentation:** 4
**Contribution:** 2
**Rating:** 8
**Confidence:** 2

**Summary:**

The authors study a continuous time population model of independent Q-Learning in stochastic games. In particular, they consider the limit in which the batch size $B \rightarrow \infty$, the number of agents $N  \rightarrow \infty$ and the update time scale $\delta t \rightarrow 0$. In this mean field limit they consider the FPE first derived by Hu et al (2019) that describes the evolution of the Q-distribution over the agents.

The main analysis concerns the variance of Q-values for which the authors derive a corresponding dynamical system and analyse its stability. This turns out to be related to the covariance between TD-targets and Q-values. In doing so, the authors are able to determine a number of sufficient conditions to guarantee consensus among agents (agreement in Q-values).

**Strengths:**

The paper is well presented and its contributions are clear. In particular, while previous works have only been able to analyse the properties of these mean-field models through numerical integration, the authors are able to derive analytic results which guarantee convergence (consensus). This is done by looking at the moments of the Q-distribution, and opens the avenue for future work to also consider properties of the mean-field model.

**Weaknesses:**

A reasonable criticism of this work is the limitation in novelty. The dynamics that are analysed are those derived by Hu et al (2019) and the main addition is the variance analysis. Despite this I do believe that this contribution is important and should be published.

Perhaps a route to improvement would be the experiment section. I appreciate that these are largely to support the theoretical results rather than building upon them, and there are space considerations to account for. Nevertheless, the authors only evaluate on the PD and EPG games and provide little exploration of their results outside of their assumptions. A useful result would be to understand the dependency of convergence on important parameters such as the exploration rate (which the authors have implicitly assumed to be 1). Most importantly, the authors should make this assumption explicit, since many previous works (e.g. Galla and Sanders: Complex dynamics in learning complicated games, Hussain et al: On the stability of learning in network games with many players) have shown that the exploration rate has a critical impact on learning dynamics. If space is a concern, the authors could reduce the space that is used in the preliminaries (we don't get to your contributions till pg 5!) or moving proofs to the appendix.

**Questions:**

N/A

---

> ### Author Response · Authors · 2025-11-29
>
> We sincerely thank the reviewer for the recognition of our contribution and the support of the theoretical research community.
>
> In this work, we formally analyze the intrinsic consensus behavior of independent Q-learning players in stochastic games from the perspective of **dynamical systems theory**. We show that the variance dynamics follow the **second-order Price equation** in evolutionary game theory and are governed by the covariance between Q-values and TD targets. For anonymous games in large populations, consensus is guaranteed.
>
> We greatly appreciate the reviewer's valuable feedback on incorporating more experiments on the sensitivity of the consensus behavior to the parameters, which strengthens the practical implications of our theoretical findings. By combining rigorous analysis with robust empirical evidence, we aim to deepen the theoretical understanding of learning stability, efficiency, and interpretability of RL algorithms, paving the way for the development of **explainable** and **cooperative AI**.

---

### Official Review · Reviewer_gKx7 · 2025-10-30

**Soundness:** 4
**Presentation:** 4
**Contribution:** 4
**Rating:** 4
**Confidence:** 3

**Summary:**

This paper investigates how the alignment of Q-values and policies across agents can emerge intrinsically in multi-agent reinforcement learning systems without explicit coordination or communication.
The authors analyze independent Q-learning in anonymous stochastic games, derive deterministic ordinary differential equations and Fokker-Planck equations governing Q-value dynamics in the large-batch limit, and connect the variance of Q-values to the second-order Price equation from evolutionary game theory.

**Strengths:**

Originality:

The paper provides a novel theoretical perspective on consensus emergence in MARL without explicit communication, addressing a long-standing question in decentralized learning.
The analogy to the Price equation is creative and bridges reinforcement learning with evolutionary game theory in a mathematically rigorous way.

Quality:

The mathematical derivations are detailed and internally consistent.
Theorems are well-motivated and supported by lemmas connecting microscopic and macroscopic levels.
The use of both ODE and Fokker-Planck formulations adds rigor and generality.


Clarity:

The paper is clearly structured, with smooth transitions from intuition to formal analysis.
Definitions and assumptions are explicitly stated.
Figures effectively illustrate convergence and empirical-theoretical agreement.

Significance:

The results enhance understanding of emergent coordination in large-scale MARL and could inspire new communication-free learning algorithms for swarm robotics or population-scale systems.
Establishing intrinsic consensus provides a theoretical complement to mean-field RL and self-play frameworks, potentially improving scalability and interpretability of MARL systems.

**Weaknesses:**

Only small-scale normal-form and two-state stochastic games are tested.

The analysis heavily relies on anonymity and large-batch limits.

Missing connection to existing convergence results.

While related work is cited, the comparison with classical convergence guarantees in independent Q-learning or mean-field MARL could be more explicit.

Although theoretically elegant, the covariance–variance mechanism is not empirically measured or visualized.

**Questions:**

How sensitive are the consensus dynamics to finite-batch noise or function approximation errors?

Could variance contraction still hold empirically when using neural function approximators?

Could local consensus emerge within sub-networks before global convergence?

Could adding controlled entropy regularization accelerate consensus or prevent premature convergence?

---

> ### Author Response · Authors · 2025-11-29
>
> We sincerely thank the reviewer for the recognition of our theoretical contribution and the insightful feedback. The following are our responses to the reviewer's questions:
>
> 1. Figure 3 shows the system's robustness to finite batch sizes. While smaller batch sizes introduce fluctuations, the contractive drift dominates the system dynamics, and the system qualitatively converges to consensus. As shown in the figure, the empirical trajectories remain strictly below the theoretical upper bound. Our theoretical derivations suggest that if the neural function approximator preserves the Lipschitz property of the target values, consensus is expected to emerge.
>
> 2. Since consensus relies on the contractive nature of the TD operator, it is expected that consensus will emerge if a neural function approximator is sufficiently smooth in the sense that it preserves the Lipschitz property of the target values.
>
> 3. In the derivations, we find that the difference of arbitrary two agents' Q-values decays to zero exponentially. Therefore, local consensus can emerge within sub-networks with smaller Q-value gaps before global consensus.
>
> 4. Entropy regularization induces an additional term in the variance dynamics that boosts the consensus behavior. For A2C, it can be shown that entropy regularization could not only promote the stability and efficiency but also promote agents' convergence to quantal response equilibria.

---

### Official Review · Reviewer_A6fL · 2025-10-31

**Soundness:** 1
**Presentation:** 3
**Contribution:** 1
**Rating:** 2
**Confidence:** 4

**Summary:**

This paper introduces an analysis of the learning dynamics of Q-learning algorithms under the assumptions of anonymity, infinite population, negligible single-agent effect, large-batch/LLN limit, and  time-scale separation with small $\alpha$. Under these assumptions, the authors are able to proof that consensus amongst Q-learning agents emerges without having to explicitly optimize for it. The theoretical contraction bounds of the paper are validated in the Prisoner's Dilemma and Ecological Public Goods environments.

**Strengths:**

The paper is well written, the math is straightforward and correct to the best of my knowledge under the established conditions and assumptions of the setup. Morever, there is some value in the clean mean-field derivation connecting ODE/FPE views of independent Q-learning in anonymous stochastic games, that this paper introduces. Although I do not think that this contribution is by itself enough to advocate for acceptance. The simple experiments empirically illustrate the claims of the paper.

**Weaknesses:**

My main issue with this paper is that the “consensus” result largely follows from very strong simplifying assumptions (anonymity, infinite population or negligible single-agent effect, large-batch/LLN limit, and often a time-scale separation with small α). Under those, each agent’s drift no longer depends on its identity and reduces to a linear contraction toward a shared fixed point. That makes consensus almost inevitable. In other words, the paper’s core claim becomes close to trivial because of the setup, not because of a new argument.

The problem with the mean field analysis done with this paper stems from **Lemma 3.1.** This lemma essentially states that “the conditional reward $E[r(i)|s, a]$ of agent $i$ when taking action $a$ in state $s$ becomes independent of the index $i$ as the population size $N \rightarrow \infty$”. In other words, the reward of any given agent does not depend on the agent’s identity. Under this condition the result of getting consensus is completely unsurprising, since agents no longer care about their own reward but instead a **shared objective**. This is equivalent to joint reward maximization, which is well known in multi-agent RL literature to converge to a welfare maximizing, colluded policy. This, in conjunction to the assumption of anonymity naturally leads to the policies of different agents also converging to the same distribution.

Having established that, it is very hard for me to find anything particularly useful or insightful in the analysis (the main contribution of this paper), which may help steering and informing more “scalable and efficient intelligent systems”.

**Minor errors/typos:**

Some important pieces of related work are missing, which also study the emergence of “collusion” in RL algorithms, particularly the dynamics of independent Q-learners for 2-player general-sum iterated games. [1, 2]

**Line 36**: The authors state “Moreover, CTDE implicitly assumes that agents are fully cooperative, which does not always hold in real-world MAS (Kong et al., 2024).” CTDE to my understanding is a much older framework that pre-dates RL, but more importantly, makes no assumptions about the competitiveness/cooperativeness of the environment.

**Line 498**: “is described” should be “are described”  .

References:

[1] Calvano, E., Calzolari, G., Denicolo, V., and Pastorello, S. Artificial intelligence, algorithmic pricing, and collusion. American Economic Review, 110(10):3267–3297, 2020b.

[2] Bertrand, Q., Duque, J., Calvano, E., and Gidel, G. (2023). Q-learners can provably collude in the iterated prisoner’s dilemma.

**Questions:**

1. How can these findings inform real multi-agent systems with finite agents, noise, and heterogeneity? More specifically, are there any insights that the authors believe could be used to develop better MARL algorithms?

2. Are there any conditions (e.g. particular game instances), where reaching consensus might actually be an undersirable outcome? What kind of methods could be used to mitigate this problem in practice?

---

> ### Author Response · Authors · 2025-11-29
>
> We genuinely thank the reviewer for the careful reading and the insightful feedback. We also appreciate the reviewer for pointing out the two valuable related works on the emergence of collusion in MARL systems. The following are our responses to the reviewer's questions, regarding the contribution of our work, and the practical implications of consensus:
>
> 1. Our experiments demonstrate that the emergence of consensus is robust in practical settings. Specifically, we show that the system qualitatively converges to consensus even with finite populations and finite batch sizes. Leveraging tools from evolutionary game theory and statistical physics (e.g., pair approximation), the conclusion can be extended to heterogeneous populations.  We also establish the link between the evolution of system variance and the **second-order Price equation** in evolutionary game theory. Moreover, although not detailed in the manuscript, we can prove that entropy regularization induces an additional negative term in the variance dynamics. In other words, entropy regularization not only encourages exploration but also further facilitates the system's convergence to consensus. Therefore, our framework also offers a deeper understanding of algorithmic behavior, which aids in parameter selection and the validation of new mechanisms.
>
> 2. Consensus is undesirable in scenarios that require anti-coordination, such as traffic merges or crowd evacuations. Collusion is also understood as a form of consensus, which is undesirable from the consumers’ perspective. Since the consensus process is dominated by the covariance between Q-values and TD targets, mechanisms that induce a higher covariance where higher Q-values are associated with higher TD errors can promote divergence among individuals.

---

### Official Review · Reviewer_hRrN · 2025-11-01

**Soundness:** 2
**Presentation:** 2
**Contribution:** 2
**Rating:** 2
**Confidence:** 3

**Summary:**

This paper studies the emergence of consensus in multi-agent Q-learning for anonymous stochastic games, where rewards and transitions depend only on aggregate behavior, not individual identities. The authors formulate a mean-field limit, derive deterministic ODEs and Fokker-Plank equations for the distribution of Q-values, and analyze the evolution of their variance. Under large-batch and large-population limits, they prove that independent learners' Q-values converge exponentially to a equilibrium.

**Strengths:**

- The emergence of consensus without explicit communication is interesting and relevant to MARL research.
- Connecting individual-level and population-level learning dynamics through ODE and Fokker-Planck formulations is interesting.

**Weaknesses:**

- The paper suffers from numerous notational inconsistencies and premature symbol use:
-- The TD error appears in (4) before being formally defined.
-- a + a^{-i} in (6) is non-standard and never defined. It appears to denote joint action concatenation.
-- \tilde{R} is inconsistently used. Lemma 2.3 defined \tilde{R}(a\mid s,x,s') but (8) uses \tilde{R}(a,s,x(a+a^{-i}),s') without \mid.
-- The notation \bar{X} alternates between a finite-N average in (9) and its N\rightarrow\infty limit in (8).
These inconsistencies make parts of the proof difficult to follow and suggest insufficient mathematical editing.

- Given that rewards and transitions are anonymous across agents, anonymous Q-values is expected. In that sense, "consensus" is an expected symmetry outcome rather than an emergent algorithmic property. For example, Lemma 2.3 abstracts the agent identity without going to the limit N\rightarrow \infty or using Q-learning dynamics. Therefore, intuitively, the long-run behavior of other learning dynamics may also not depend on agent identity as well. However, this contrasts with consensus-protocol works (Kar et al, 2013; Zhang et al, 2018) that explicitly enforce alignment of distributed estimates corresponding to the sum of local objectives. The paper should clearly delineate that its result shows identity-independent convergence, not consensus on a shared global value function. Furthermore, the title should highlight anonymous stochastic games.

- The authors switch to a continuous-time ODE but not tie the continuous-time dynamics back to the discrete Q-learning recursion in (4).

**Questions:**

- How does your notion of consensus compare, both mathematically and operationally, with the consensus-innovations frameworks (Kar 2013; Zhang 2018)?

---

> ### Author Response · Authors · 2025-11-29
>
> We appreciate the reviewer for the valuable comment. In this work we study consensus that arises from independent Q-learners in large-scale anonymous stochastic games **without** any explicit coordination and communication mechanisms. We construct the dynamic equation for the evolution of system variance and show that it closely follows the **second-order Price equation** in evolutionary game theory, with the consensus rate governed by the covariance between Q-values and TD targets. Establishing consensus thus reduces to proving that the variance of Q-values across the population decays to zero. These results and principles apply to **general stochastic games**.
>
> On the other hand, the consensus-innovations frameworks, such as those by Kar et al. (2013) and Zhang et al. (2018) rely on explicit algorithmic protocols to enforce agreement. To prove consensus, they utilize spectral graph theory and stochastic approximation theory to demonstrate that the deviation of local estimates from the average almost surely converges to zero.

---

### Meta-Review · Area_Chair_NHfv · 2026-01-07

**Summary:**

The paper studies emergence of consensus between Q-learning agents in multi-agent stochastic games, where each agent is independent and anonymous. All reviewers recognized that demonstration of emergence of consensus without explicit communication is an interesting contribution. However, the overall sentiment of the reviews was negative due to multiple limitations and weaknesses pointed out in the reviewers. The authors' responses to each of the reviews during the rebuttal were quite concise and not addressing any of critical points raised by the reviewers (my impression of the author's responses as if they were auto-generated).

Due multiple unaddressed weaknesses, I recommend rejecting the paper as not ready for publication in the current form.

**Reviewer Concerns:**

- Triviality of the paper's claims under the made assumptions (as pointed by reviewers hRrN, A6fL, gKx7).
  - In particular, to quote A6fL: `My main issue with this paper is that the “consensus” result largely follows from very strong simplifying assumptions (anonymity, infinite population or negligible single-agent effect, large-batch/LLN limit, and often a time-scale separation with small α). Under those, each agent’s drift no longer depends on its identity and reduces to a linear contraction toward a shared fixed point. That makes consensus almost inevitable. In other words, the paper’s core claim becomes close to trivial because of the setup, not because of a new argument.`
- Missing connection to related work on classical convergence results in independent Q-learning or mean-field MARL (as pointed by gKx7)
- Limited empirical validation and limited analysis of the empirical results (as pointed by gKx7)
- Confusing / inconsistent writing style + multiple typos (pointed out by multiple reviewers)

**Reviewer Scores:**

- My best assessment is that all reviewers would've likely maintained their scores given the authors' very brief responses.
- Reviewer aX3W could've lowered their score of 8 given the weaknesses identified by other reviewers.

---

### Decision · Program_Chairs · 2026-01-26

Reject